# Relationship between evacuation after the Great East Japan Earthquake and new-onset hyperuricemia: A 7-year prospective longitudinal study of the Fukushima Health Management Survey

Kazuya Honda[1,2⦿], Kanako Okazaki[3,4,5⦿], Kenichi Tanaka[1,6]*, Eri Kobari[1‡], Sakumi Kazama[3,6‡], Shigeatsu Hashimoto[2,3‡], Tetsuya Ohira[3,5⦿], Akira Sakai[3,7‡], Seiji Yasumura[3,8‡], Masaharu Maeda[3,9‡], Hirooki Yabe[3,10‡], Mitsuaki Hosoya[3,11‡], Atsushi Takahashi[3,12‡], Mayumi Harigane[3‡], Hironori Nakano[3,5‡], Fumikazu Hayashi[3,5‡], Masanori Nagao[3,5‡], Michio Shimabukuro[3,13‡], Hitoshi Ohto[3‡], Kenji Kamiya[3,14‡], Junichiro J. Kazama[1,3,6‡]

1 Department of Nephrology and Hypertension, Fukushima Medical University School of Medicine, Fukushima, Japan, 2 Department of Endocrinology, Metabolism, Diabetology, and Nephrology, Fukushima Medical University Aizu Medical Center, Aizuwakamatsu, Japan, 3 Radiation Medical Science Center for the Fukushima Health Management Survey, Fukushima Medical University, Fukushima, Japan, 4 Department of Physical Therapy, Fukushima Medical University School of Health Sciences, Fukushima, Japan, 5 Department of Epidemiology, Fukushima Medical University School of Medicine, Fukushima, Japan, 6 Division of Advanced Community Based Care for Lifestyle Related Diseases, Fukushima Medical University School of Medicine, Fukushima, Japan, 7 Department of Radiation Life Sciences, Fukushima Medical University School of Medicine, Fukushima, Japan, 8 Department of Public Health, Fukushima Medical University School of Medicine, Fukushima, Japan, 9 Department of Disaster Psychiatry, Fukushima Medical University School of Medicine, Fukushima, Japan, 10 Department of Neuropsychiatry, Fukushima Medical University School of Medicine, Fukushima, Japan, 11 Department of Pediatrics, Fukushima Medical University School of Medicine, Fukushima, Japan, 12 Department of Gastroenterology, Fukushima Medical University School of Medicine, Fukushima, Japan, 13 Department of Diabetes, Endocrinology and Metabolism, Fukushima Medical University School of Medicine, Fukushima, Japan, 14 Research Institute for Radiation Biology and Medicine, Hiroshima University, Hiroshima, Japan

⦿ These authors contributed equally to this work.
‡ EK, SK, SH, AS, SY, MM, HY, MH, AT, MH, HN, FH, MN, MS, HO, KK and JJK also contributed equally to this work.
* kennichi@fmu.ac.jp

## Abstract

## Introduction

On March 11, 2011, the Great East Japan Earthquake occurred in Japan, with a nuclear accident subsequently occurring at the Fukushima Daiichi Nuclear Power Plant. The disaster forced many evacuees to change particular aspects of their lifestyles. However, the effect of evacuation on the new-onset of hyperuricemia have not been sufficiently elucidated. This study assessed the association between evacuation and new-onset hyperuricemia after the earthquake based on the Fukushima Health Management Survey from a lifestyle and socio-psychological perspective.

**Data Availability Statement:** Data underlying the findings in this study cannot be made publicly available due the nature of ethical approval for the study. Interested researchers may submit requests to the Fukushima Medical University's Ethics Committee (Contact information: Email: rs@fmu. ac.jp) for access to confidential data.

**Funding:** This work was supported by the National Health Fund for Children and Adults Affected by the Nuclear Incident. No financial support has been received from any source other than this fund. There was no additional external funding received for this study. The funders had no role in study design, data collection and analysis, decision to publish, or preparation of the manuscript.

**Competing interests:** The authors have declared that no competing interests exist.

## Materials and methods

This is a 7-year prospective longitudinal study included 18,140 residents (6,961 men and 11,179 women) with non-hyperuricemia who underwent both the Comprehensive Health Check and the Mental Health and Lifestyle Survey in fiscal year 2011. Associations between new-onset hyperuricemia and lifestyle- and disaster-related factors, including evacuation, were estimated using a Cox proportional hazards regression model analysis. Hyperuricemia was defined as uric acid levels > 7.0 mg/dL for men and > 6.0 mg/dL for women.

## Results

During a median follow-up of 4.3 years, 2,996 participants (1,608 men, 23.1%, 1,388 women, 12.4%) newly developed hyperuricemia. Significant associations were observed between evacuation and onset of hyperuricemia in women (adjusted hazard ratio 1.18, 95% confidence interval, 1.05–1.32, p = 0.007), but not in men (adjusted hazard ratio 1.11, 95% confidence interval, 0.99–1.24, p = 0.067).

## Discussion

Evacuation after a natural disaster is an independent risk factor for the new-onset of hyperuricemia in women. The possibility of hyperuricemia developing in response to natural disasters should be considered.

## Introduction

Uric acid is an oxidative end product of purine metabolism and a potent antioxidant. In humans, about half of the antioxidants in plasma are uric acid [1]. Therefore, uric acid is an essential substance for humans. In fact, there is a certain correlation between uric acid levels and lifespan in primates [2]. However, hyperuricemia has been reported to increase the risk of end-stage renal failure [3], cardiovascular disease, and death [4–8]. In addition to hypertension [9–11], hyperuricemia is also associated with obesity, diabetes mellitus, and dyslipidemia [12], a condition deeply related to lifestyle habits, such as alcohol consumption and diet.

On March 11, 2011, the Great East Japan Earthquake, registering 9.0 on the moment magnitude scale, occurred in Japan. The earthquake provoked a tsunami that struck the Fukushima Daiichi Nuclear Power Plant, resulting in more than 160,000 Fukushima Prefecture residents being evacuated to other parts of Japan as well as abroad.

Evacuation immediately following a major disaster leads to significant lifestyle changes, particularly changes in housing, family structure, and employment; it also causes stress, insomnia, and decreased physical activity [13–15]. These changes can result in increased body weight, which is associated with an increase in lifestyle-related diseases and hyperuricemia [16–20]. Recently, we reported a cross-sectional study that suggested that evacuation after the Great East Japan Earthquake had significant and positive associations with uric acid levels both in men and women [21]. However, to date, no study has analyzed the effect of evacuation on the new-onset of hyperuricemia adjusting for physical and socio-psychological factors, and it is still unclear whether evacuation after an earthquake is related to the increased risk for hyperuricemia, in spite the fact that hyperuricemia could be a crucial comorbidity associated with mortality, cardiovascular disease, and end-stage kidney disease. Therefore, revealing the

relationship of evacuation to hyperuricemia is needed in health assessment for the victims after a natural disaster. The purpose of the present study was to validate the relationship between evacuation and new-onset hyperuricemia after a disaster using long-term longitudinal data of a 7-year follow-up in the Fukushima Health Management Survey (FHMS).

## Materials and methods

### Study population

The subjects of this study were Japanese men and women living near the Fukushima Daiichi Nuclear Power Plant in Fukushima prefecture, in communities including Tamura City (2010 Census population, 42,085), Minami-Soma City (71,661), Kawamata-machi (16,065), Hirono-machi (5495), Naraha-machi (7927), Tomioka-machi (15,854), Kawauchi-mura (3074), Okuma-machi (11,553), Futaba-machi (7171), Namie-machi (21,551), Katsurao-mura (1582), Iitate-mura (6584) and Date City (67,684), with a total 2010 population of 278,286. After the disaster, the government designated the 20-km radius around the Fukushima Daiichi Nuclear Power Plant a restricted area requiring compulsory evacuation. The government subsequently designated the 20- to 30-km area around the plant as an evacuation-prepared area in case of emergency, and areas near the 30-km radius where high-level radiation exposure was expected (>20 mSv/y) as deliberate evacuation areas (evacuation over a period of roughly one month). As a result, all residents of Hirono-machi, Naraha-machi, Tomioka-machi, Kawauchi-mura, Okuma-machi, Futaba-machi, Namie-machi, Katsurao-mura, and Iitate-mura evacuated from their homes; this was also the case for some areas of Tamura City, Minami-Soma City, Kawa-mata-machi, and Date City.

In this study, a total of 123,314 residents (62,161 men and 61,153 women) between 40 and 74 years old were registered from the above areas in fiscal year 2011. A total of 33,559 participants (14,135 men and 19,424 women) aged 40 to 74 years completed the Comprehensive Health Check of the FHMS in fiscal year 2011. Among them, 9,502 participants who did not respond to the Mental Health and Lifestyle Survey and nine participants without serum uric acid data were excluded. Moreover, those who met the diagnosis of hyperuricemia at baseline, those who did not undergo the Comprehensive Health Check of the FHMS from fiscal years 2012 to 2017, and participants without evacuation status data were excluded (5,908 participants). Thus, 18,140 participants (6,961 men and 11,179 women) were included in the final analysis (Fig 1). This corresponded to approximately 14.7% of the population between 40 and 74 years of age who were registered in the above areas in fiscal year 2011 (approximately 11.2% of men and 18.3% of women). The Comprehensive Health Check, which was conducted by the FHMS, evaluated subjective symptoms, family history, smoking and drinking history, and laboratory findings, such as blood counts, liver function, kidney function, and lipids. Detailed descriptions of the survey methods have been summarized in previous literature [13]. The Mental Health and Lifestyle Survey was conducted by the FHMS to evaluate changes in mental status and living conditions after the disaster.

This study protocol was approved by the Ethics Committee of the Fukushima Medical University School of Medicine (approval numbers 1319, 2020–239, 29064) and conformed to the ethical guidelines of the 1975 Declaration of Helsinki. The need for consent from participant was waived by Ethics Committee by obtaining consent from community representatives to conduct an epidemiological study based on the guidelines of the Council for International Organizations of Medical Science [22].

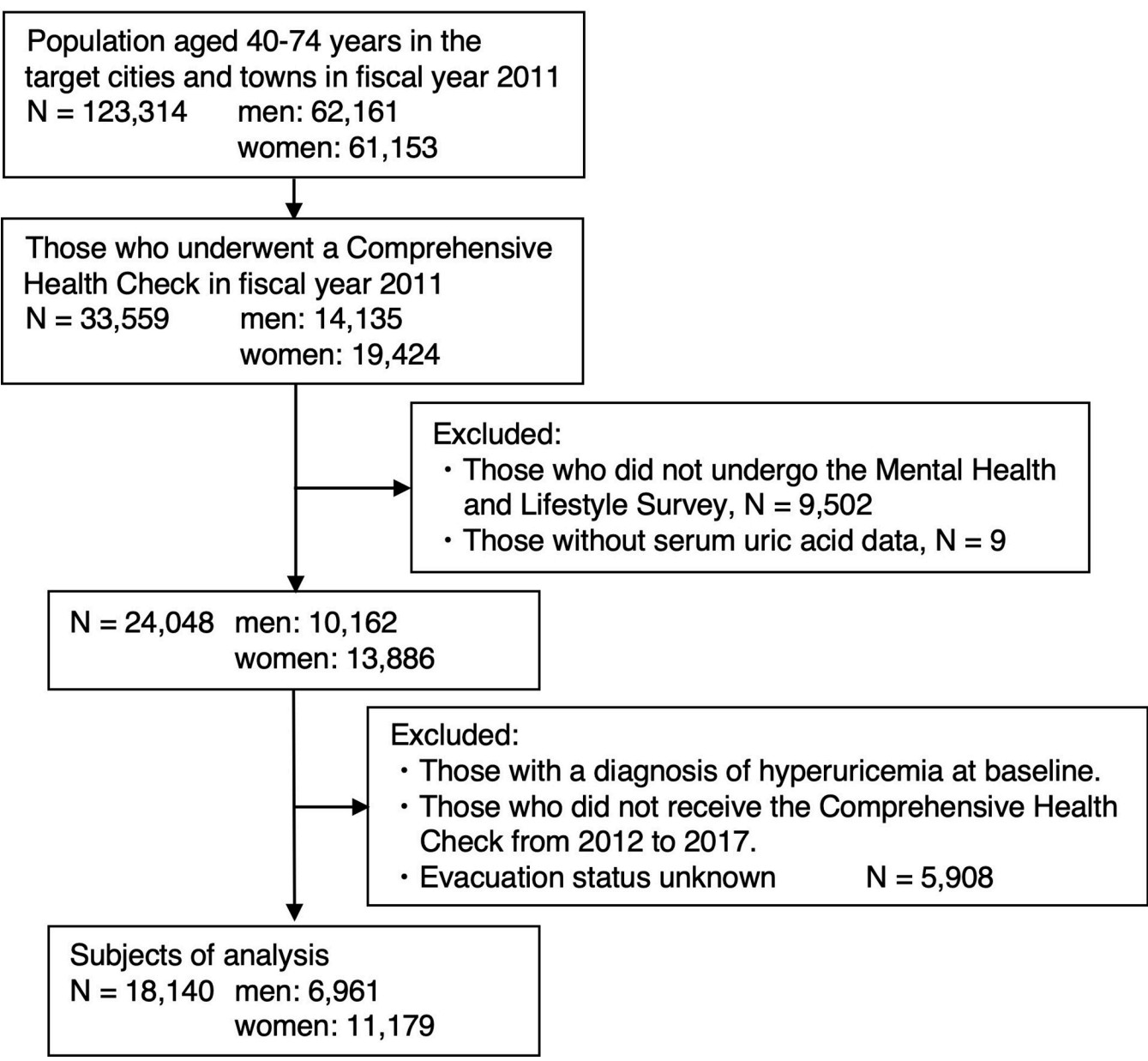

**Fig 1. Consort diagram displaying how individuals were recruited and studied during the trial.**

## Study design

This was a prospective longitudinal study. The primary outcome was new-onset of hyperuricemia defined as serum uric acid levels > 7.0 mg/dL for men and > 6.0 mg/dL for women, respectively. Those who did not have hyperuricemia as of fiscal year 2011 were followed until fiscal year 2017 to examine the factors associated with new-onset hyperuricemia, using multivariate analyses based on the results of medical examinations and interview items. In particular, we examined whether "evacuation" poses a significant risk of new-onset hyperuricemia, even after adjusting for multiple models. Due to the large sex-related differences in uric acid levels, the analysis was conducted separately for men and women.

## Measurements

Sex, age at the beginning of fiscal year (April 1, 2011), height, body weight, body mass index (BMI), systolic blood pressure, diastolic blood pressure, use of antihypertensive medication, hemoglobin A1c (HbA1c), fasting blood glucose, use of hypoglycemic agents, high-density lipoprotein cholesterol (HDL-C), low-density lipoprotein cholesterol (LDL-C), triglycerides, use of drugs for dyslipidemia, aspartate aminotransferase (AST), alanine aminotransferase (ALT), γ-glutamyl transpeptidase (γ-GT), uric acid, and serum creatinine were measured during the Comprehensive Health Check that was conducted in 13 municipalities in the evacuation zone. Body weight (kg) and height were measured without shoes and in light clothing. BMI was calculated by dividing body weight (kg) by height squared ($m^2$). Obesity was defined as a BMI $\geq$ 25.0 kg/$m^2$. Hypertension was defined as systolic blood pressure $\geq$ 140 mmHg and/or diastolic blood pressure $\geq$ 90 mmHg, or being treated for hypertension. Diabetes was defined as fasting blood glucose $\geq$ 126 mg/dL or random blood glucose $\geq$ 200 mg/dL, and/or hemoglobin A1c (HbA1c) $\geq$ 6.5%, or being treated for diabetes. Dyslipidemia was defined as low-density lipoprotein cholesterol (LDL-C) $\geq$ 140 mg/dL, fasting triglycerides $\geq$ 150 mg/dL, and/or high-density lipoprotein cholesterol (HDL-C) $<$ 40 mg/dL, or being treated for dyslipidemia. Abnormal liver function was defined as AST $\geq$ 31 U/L, ALT $\geq$ 31 U/L, or γ-GT $\geq$ 51 U/L. Abnormal renal function was defined as an estimated glomerular filtration rate (eGFR) of $<$ 60 mL/min/1.73 $m^2$ or urine protein $\geq$ +1. Since hyperuricemia has different cutoff values for men and women according to previous literature [23–25], hyperuricemia was defined as serum uric acid levels $>$ 7.0 mg/dL for men and $>$ 6.0 mg/dL for women, respectively.

In the Mental Health and Lifestyle Survey, the Japanese versions of the Kessler 6-item scale (K6) [26] and Post-traumatic Stress Disorder Checklist (PCL) [15] were used to assess the participants' mental health. The K6 consists of six brief questions about depressive and anxiety symptoms during the past 30 days, with overall scores ranging from 0 to 24. We defined psychological distress as corresponding to a K6 score $\geq$ 13 [27]. The PCL is a tool used to evaluate symptoms of post-traumatic stress disorder (PTSD) during the past 30 days. The PCL consists of 17 items, and the overall score ranges from 17 to 85. We classified participants as having probable PTSD if their overall PCL score was $\geq$ 44 [15]. Detailed information and their reliability were described in previous studies [15, 26, 27]. In addition to the K6 and PCL, data on medical history and various lifestyle factors, such as cigarette smoking status, drinking status, physical activity, sleep satisfaction, and changes in work situation were obtained from other questionnaires. Changes in work situation included unemployment or a job change after the accident. Furthermore, because this study was disaster-related, we assessed the presence of changes in residence and evacuation, and the subjects' experiences of the tsunami, as well as their experiences of the nuclear accident.

## Definitions of evacuation, evacuees, and non-evacuees

Evacuation was defined as an experience of moving into shelter or temporary housing at the time of the survey. The term "evacuees" were defined as residents of the thirteen municipalities whose entire area was evacuated or those with a self-reported experience of moving into shelters or temporary housing.

Non-evacuees were defined as the residents of the four municipalities whose residence at the time of the survey was not a shelter or temporary housing. However, among the non-evacuation categories, those whose residence at the time of the survey was a shelter or temporary housing were defined as evacuees.

## Statistical analysis

We first conducted t-tests for continuous variables and $\chi^2$ tests for categorical variables to perform comparisons between evacuees and non-evacuees for analysis of the characteristics of the target population who did not have hyperuricemia at baseline.

Next, as the cutoff values for hyperuricemia were different between men and women and the clinical impact of hyperuricemia might also be different, Cox proportional hazards regression model analyses with the onset of hyperuricemia as the dependent variable were performed separately for men and women. First, referring to previous cross-sectional studies [21], we adjusted for the independent variables of age, evacuation, BMI, systolic blood pressure, fasting blood glucose, triglycerides, eGFR, smoking status, drinking status, and unemployment in Model 1. Next, in order to consider physical and socio-psychological factors, in addition to Model 1, sleep dissatisfaction, physical activity, tsunami experience, nuclear accident experience, and PTSD were adjusted as independent variables for Model 2. Since HDL-C, LDL-C, and triglycerides statistically strongly correlate with each other, triglyceride level was adopted as representative of dyslipidemia, as in previous reports [28]. Since there was a strong correlation between K6 score and PCL score, PTSD (PCL score $\geq$ 44) was adopted as an independent factor in this study.

In the present study, the dependent variable excluded missing data, while the explanatory variables included missing data. Specifically, dummy variables were created for missing and non-missing values, with missing values set to "1" and non-missing values set to "0" and used in the multivariate analysis.

All analyses were performed using SAS version 9.4 (SAS Institute, Cary, NC, USA). A two-tailed test was performed, with a significance level of $p < 0.05$.

## Results

A total of 18,140 patients, including 6,961 men and 11,179 women, met the conditions of this study (Fig 1).

Table 1 shows the results of the Comprehensive Health Check for the target group that did not have hyperuricemia at baseline between evacuees and non-evacuees stratified by sex. In men, the evacuee group was significantly younger, had a higher BMI, lower blood pressure, higher triglycerides, more liver dysfunction, higher eGFR, and higher uric acid levels than the non-evacuee group. In women, in addition to the above, the evacuee group had significantly lower fasting blood glucose levels than the non-evacuee group. The results of the Mental Health and Lifestyle Survey at baseline between evacuees and non-evacuees stratified by sex are shown in Table 2. In men, there were significant differences in smoking and drinking status, sleep satisfaction, changes in work situation, unemployment experience, nuclear accident experience, psychological distress, and PTSD between the evacuee and non-evacuee groups. In women, in addition to the above, there were significant differences in physical activity between the evacuee and non-evacuee groups.

The results of the Comprehensive Health Check for the target group that did not have hyperuricemia at baseline stratified by sex and the development of hyperuricemia during the follow-up period is shown in S1 Table. During a median follow-up of 4.3 years, 1,608 (23.1%) men and 1,388 (12.4%) women newly met the criteria for hyperuricemia. The prevalence of diabetes in the hyperuricemia group was significantly lower in men but higher in women as compared to that in the non-hyperuricemia group. The prevalence of obesity, hypertension, dyslipidemia, abnormal liver function, and abnormal renal function were higher in the hyperuricemia group than in the non-hyperuricemia group in both men and women. The results of the Mental Health and Lifestyle Survey at baseline stratified by sex and the development of

**Table 1. Clinical and biochemical characteristics of the 18,140 participants stratified by sex and evacuation status.**

| | Men | | | | Women | | | |
|---|---|---|---|---|---|---|---|---|
| | **Total** | **Evacuees** | **Non-evacuees** | **p** | **Total** | **Evacuees** | **Non-evacuees** | **p** |
| n (%) | 6,961 | 3,231 (46.4) | 3,730 (53.6) | | 11,179 | 5,174 (46.3) | 6,005 (53.7) | |
| Age (years) | 61.2 (8.6) | 60.8 (8.5) | 61.5 (8.6) | 0.001 | 59.2 (9.1) | 58.6 (9.1) | 59.7 (9.0) | <0.001 |
| BMI (kg/m$^2$) | 24.3 (3.0) | 24.5 (3.1) | 24.2 (3.0) | 0.001 | 23.4 (3.5) | 23.5 (3.6) | 23.2 (3.4) | <0.001 |
| BMI ≥ 25 kg/m$^2$ (%) | 39.2 | 41.4 | 37.2 | <0.001 | 28.5 | 30.2 | 27.1 | <0.001 |
| Systolic blood pressure (mmHg) | 133 (15) | 132.6 (15) | 133.4 (15) | 0.037 | 129 (16) | 128 (16) | 129 (16) | <0.001 |
| Diastolic blood pressure (mmHg) | 81 (10) | 80.6 (10) | 81.1 (10) | 0.035 | 77 (10) | 77 (10) | 78 (10) | <0.001 |
| Hypertension (%) | 57.3 | 57.0 | 57.5 | 0.718 | 43.3 | 42.5 | 43.9 | 0.136 |
| HbA1c (%) | 5.6 (0.8) | 5.6 (0.9) | 5.6 (0.8) | 0.805 | 5.5 (0.6) | 5.5 (0.7) | 5.5 (0.6) | 0.085 |
| Fasting blood glucose (mg/dL) | 100 (93–111) | 100 (93–111) | 100 (93–110) | 0.678 | 94 (89–102) | 94 (88–102) | 95 (89–102) | 0.013 |
| Diabetes (%) | 17.2 | 17.7 | 16.7 | 0.255 | 7.7 | 7.9 | 7.5 | 0.396 |
| HDL-C (mg/dL) | 56 (14) | 56 (15) | 56 (14) | 0.159 | 64 (15) | 64 (15) | 64 (15) | 0.051 |
| LDL-C (mg/dL) | 123 (32) | 124 (33) | 123 (32) | 0.553 | 130 (32) | 131 (33) | 130 (31) | 0.209 |
| Triglycerides (mg/dL) | 107 (76–152) | 109 (78–157) | 104 (74–148) | <0.001 | 90 (66–126) | 93 (67–128) | 89 (65–124) | <0.001 |
| Dyslipidemia (%) | 57.0 | 58.1 | 56.0 | 0.080 | 58.4 | 57.4 | 59.2 | 0.046 |
| AST (U/L) | 24 (20–29) | 24 (20–30) | 24 (20–29) | <0.001 | 21 (18–25) | 21 (18–26) | 21 (18–25) | 0.754 |
| ALT (U/L) | 22 (16–32) | 23 (17–34) | 22 (16–30) | <0.001 | 17 (13–23) | 17 (13–24) | 16 (13–22) | 0.001 |
| γ-GT (U/L) | 33 (22–56) | 35 (24–59) | 32 (21–53) | <0.001 | 18 (14–27) | 19 (14–29) | 18 (13–26) | <0.001 |
| Abnormal liver function (%) | 44.9 | 47.9 | 42.2 | <0.001 | 18.8 | 20.6 | 17.2 | <0.001 |
| eGFR (mL/min/1.73 m$^2$) | 73.9 (13.4) | 74.9 (13.2) | 73.1 (13.5) | <0.001 | 74.6 (12.7) | 75.7 (13.2) | 73.6 (12.3) | <0.001 |
| Abnormal renal function (%) | 13.0 | 12.6 | 13.3 | 0.338 | 10.5 | 9.1 | 11.6 | <0.001 |
| Uric acid (mg/dL) | 5.5 (1.0) | 5.5 (1.0) | 5.4 (1.0) | 0.040 | 4.3 (0.9) | 4.3 (0.9) | 4.2 (0.9) | 0.007 |

The values in the table indicate the average value (standard deviation) or the percentage. Fasting blood glucose, triglycerides, AST, ALT and γ-GT for which a normal distribution could not be confirmed are reported as the median (25–75% percentile). BMI: body mass index, HbA1c: hemoglobin A1c, HDL-C: high-density lipoprotein cholesterol, LDL-C: low-density lipoprotein cholesterol, AST: aspartate aminotransferase, ALT: alanine aminotransferase, γ-GT: γ-glutamyl transpeptidase, eGFR: estimated glomerular filtration rate.

hyperuricemia is shown in S2 Table. The percentage of participants who experienced evacuation after the earthquake was significantly higher in the hyperuricemia group than in the non-hyperuricemia group in both men and women. Smoking and drinking status differed significantly between the hyperuricemia and non-hyperuricemia groups in both men and women. The percentage of participants who experienced changes in work situation and unemployment experience were significantly higher in the hyperuricemia group than in the non-hyperuricemia group only in men.

The results of the association between the new-onset of hyperuricemia (> 7.0 mg/dL) and earthquake-related items and lifestyle diseases in men are shown in Table 3. Age, BMI, systolic blood pressure, fasting blood glucose, triglycerides, eGFR, smoking status, and drinking status had significant effects on the risk of new-onset of hyperuricemia after multivariable-adjustment (Model 1 and 2). Unemployment experience had a significant effect on the risk of new-onset hyperuricemia in univariate analysis, but not in multivariate models. Among men, the incidences of hyperuricemia were 52.2 and 63.4 per 1,000 person-years in non-evacuees and evacuees, respectively. Evacuation had a significant effect on the risk of the new-onset of hyperuricemia in univariate analysis (hazard ratio 1.19, 95% confidence interval 1.08–1.31, p = 0.001), but not in multivariate models (Model 1 and 2) in men.

The results of the association between the new-onset of hyperuricemia (> 6.0 mg/dL) and earthquake-related items and lifestyle diseases in women are shown in Table 4. Evacuation,

**Table 2. Lifestyle characteristics of the 18,140 participants stratified by sex and evacuation status.**

| | | Men | | | | Women | | | |
|---|---|---|---|---|---|---|---|---|---|
| | | Total | Evacuees | Non-evacuees | p | Total | Evacuees | Non-evacuees | p |
| Smoking status | Never smoker | 1,829 (26.6) | 807 (25.3) | 1,022 (27.7) | <0.001 | 9,432 (88.0) | 4,233 (85.4) | 5,199 (90.2) | <0.001 |
| | Quit smoking | 3,176 (46.1) | 1,441 (45.2) | 1,735 (47.0) | | 638 (6.0) | 341 (6.9) | 297 (5.2) | |
| | Current smoker | 1,879 (27.3) | 942 (29.5) | 937 (25.4) | | 750 (6.1) | 385 (7.8) | 365 (4.6) | |
| Drinking status | Never drinker | 1,687 (24.4) | 774 (24.1) | 913 (24.6) | 0.004 | 7,652 (70.3) | 3,435 (68.3) | 4,217 (72.0) | <0.001 |
| | Quit drinking | 316 (4.6) | 152 (4.7) | 164 (4.4) | | 113 (1.0) | 59 (1.2) | 54 (0.9) | |
| | <44 g/day | 3,278 (47.3) | 1,463 (45.6) | 1,815 (48.9) | | 2,902 (26.7) | 1,409 (28.0) | 1,493 (25.5) | |
| | ≥ 44 g/day | 1,645 (23.8) | 822 (25.6) | 823 (22.2) | | 220 (2.0) | 130 (2.6) | 90 (1.5) | |
| Sleep satisfaction | Satisfied | 2,368 (41.1) | 890 (33.6) | 1,478 (47.5) | <0.001 | 2,475 (27.0) | 870 (20.5) | 1,605 (32.5) | <0.001 |
| | Slightly dissatisfied | 2,430 (42.2) | 1,177 (44.4) | 1,253 (40.3) | | 4,524 (49.3) | 2,099 (49.6) | 2,425 (49.1) | |
| | Very dissatisfied | 766 (13.3) | 449 (16.9) | 317 (10.2) | | 1,673 (18.2) | 947 (22.4) | 726 (14.7) | |
| | Unable to sleep | 198 (3.4) | 135 (5.1) | 63 (2.0) | | 508 (5.5) | 320 (7.6) | 188 (3.8) | |
| Physical activity | Every day | 1,343 (19.6) | 619 (19.5) | 724 (19.7) | 0.371 | 1,559 (14.2) | 733 (14.4) | 826 (14.1) | 0.002 |
| | 2–4 times a week | 1,587 (23.2) | 747 (23.6) | 840 (22.9) | | 2,722 (24.8) | 1,331 (26.2) | 1,391 (23.7) | |
| | Once a week | 1,040 (15.2) | 457 (14.4) | 583 (15.9) | | 1,667 (15.2) | 719 (14.1) | 948 (16.2) | |
| | None | 2,868 (41.9) | 1,345 (42.5) | 1,523 (41.5) | | 5,011 (45.7) | 2,307 (45.3) | 2,704 (46.1) | |
| Changes in work situation | Yes | 3,953 (58.4) | 2,239 (71.5) | 1,714 (47.2) | <0.001 | 5,964 (57.1) | 3,383 (70.0) | 2,581 (45.9) | <0.001 |
| Unemployment experience | Yes | 1,407 (247) | 996 (37.1) | 411 (13.6) | <0.001 | 2,680 (28.2) | 1,871 (42.2) | 809 (16.0) | <0.001 |
| Tsunami experience | Yes | 1,616 (23.2) | 766 (23.7) | 850 (22.8) | 0.365 | 1,896 (17.0) | 883 (17.1) | 1,013 (16.9) | 0.782 |
| Nuclear accident experience | Yes | 3,924 (56.4) | 2,073 (64.2) | 1,851 (49.6) | <0.001 | 5,942 (53.2) | 3,085 (59.6) | 2,857 (47.6) | <0.001 |
| Psychological distress | K6 ≥ 13 | 717 (11.4) | 445 (15.5) | 272 (8.0) | <0.001 | 1,755 (17.4) | 982 (21.1) | 773 (14.2) | <0.001 |
| PTSD | PCL ≥ 44 | 1,169 (18.3) | 666 (22.8) | 503 (14.6) | <0.001 | 2,536 (24.7) | 1,370 (29.0) | 1,166 (21.1) | <0.001 |

The values in the table indicate the number (percentage). K6: Kessler 6-item scale, PTSD: Post-traumatic stress disorder, PCL: Post-traumatic Stress Disorder Checklist. Various lifestyle factors, such as cigarette smoking status, drinking status, physical activity, sleep satisfaction, changes in work situation, and unemployment experience were obtained from other questionnaires. Because this study was disaster-related, the subjects' experiences of the tsunami and the nuclear accident were assessed as well. Psychological distress and PTSD were defined as corresponding to a K6 score ≥ 13 and PCL score ≥ 44, respectively.

BMI, systolic blood pressure, triglycerides, eGFR, smoking status, and drinking status had significant effects on the risk of new-onset hyperuricemia after multivariate-adjustment in women (Model 1 and 2). Among women, the incidences of hyperuricemia were 24.6 and 31.2 per 1,000 person-years in non-evacuees and evacuees, respectively. The adjusted hazard ratio of evacuation for the new-onset of hyperuricemia was 1.18 (95% confidence interval 1.05–1.32, p = 0.007, Model 2).

We analyzed the hazard ratios of evacuation for hyperuricemia according to the baseline variates by sex in S3 and S4 Tables. In men, evacuation was shown to be a significant risk factor for the new-onset of hyperuricemia in drinkers (including those who quit, as well as current drinkers) and subjects on anti-hypertensive medication (S3 Table). In women, evacuation was shown to be a significant risk factor for the new-onset of hyperuricemia in subjects aged < 65 years, BMI < 25.0 kg/m$^2$, never smokers, and those not on anti-hypertensive agents and with no diabetes or abnormal renal function (S4 Table).

## Discussion

In this prospective longitudinal study, disaster evacuation was found to be one of the risk factors for new-onset hyperuricemia in women but not in men. This study is extremely valuable because there have been no previous longitudinal reports of the association between hyperuricemia and psychological and social influences, such as evacuation due to disasters.

**Table 3. Cox regression analyses of the association between hyperuricemia (levels higher than 7 mg/dL) and earthquake-related items and lifestyle-related factors in 6,961 men.**

| | Reference | Univariate analysis | | Model 1 | | Model 2 | |
|---|---|---|---|---|---|---|---|
| | | HR (95% CI) | p | HR (95% CI) | p | HR (95% CI) | p |
| Age (years), *per 1 SD increase* | | 0.99 (0.94–1.04) | 0.756 | 0.92 (0.86–0.99) | 0.029 | 0.91 (0.84–0.98) | 0.011 |
| Evacuation status | *Non-evacuee* | | | | | | |
| Evacuee | | 1.19 (1.08–1.31) | 0.001 | 1.11 (1.00–1.24) | 0.051 | 1.11 (0.99–1.24) | 0.065 |
| BMI (kg/m$^2$), *per 1 SD increase* | | 1.21 (1.16–1.27) | <0.001 | 1.15 (1.09–1.22) | <0.001 | 1.15 (1.09–1.22) | <0.001 |
| Systolic blood pressure (mmHg), *per 1 SD increase* | | 1.21 (1.15–1.27) | <0.001 | 1.17 (1.11–1.24) | <0.001 | 1.17 (1.11–1.24) | <0.001 |
| Fasting blood glucose (mg/dL), *per 1 SD increase* | | 0.97 (0.91–1.02) | 0.208 | 0.93 (0.87–0.98) | 0.012 | 0.93 (0.87–0.98) | 0.012 |
| HDL-C (mg/dL), *per 1 SD decrease* | | 1.11 (1.05–1.16) | <0.001 | | | | |
| LDL-C (mg/dL), *per 1 SD increase* | | 0.92 (0.87–0.96) | <0.001 | | | | |
| Triglycerides (mg/dL), *per 1 SD increase* | | 1.13 (1.10–1.17) | <0.001 | 1.09 (1.05–1.13) | <0.001 | 1.09 (1.05–1.13) | <0.001 |
| eGFR (mL/min/1.73 m$^2$), *per 1 SD decrease* | | 1.27 (1.20–1.34) | <0.001 | 1.33 (1.26–1.42) | <0.001 | 1.33 (1.26–1.42) | <0.001 |
| Smoking status | *Never smoker* | | | | | | |
| Quit smoking | | 1.23 (1.09–1.39) | 0.001 | 1.14 (1.00–1.30) | 0.055 | 1.14 (0.99–1.30) | 0.061 |
| Current smoker | | 1.29 (1.12–1.48) | <0.001 | 1.37 (1.18–1.59) | <0.001 | 1.38 (1.18–1.60) | <0.001 |
| Drinking status | *Never drinker* | | | | | | |
| Quit drinking | | 1.10 (0.84–1.44) | 0.490 | 1.00 (0.74–1.34) | 0.997 | 1.00 (0.75–1.35) | 0.985 |
| <44 g/day | | 1.24 (1.08–1.41) | 0.002 | 1.28 (1.11–1.47) | <0.001 | 1.28 (1.11–1.48) | <0.001 |
| ≥ 44 g/day | | 1.72 (1.49–1.98) | <0.001 | 1.73 (1.48–2.02) | <0.001 | 1.74 (1.48–2.03) | <0.001 |
| Unemployment experience | *No* | | | | | | |
| Yes | | 1.17 (1.04–1.32) | 0.012 | 1.14 (1.00–1.30) | 0.058 | 1.14 (0.99–1.30) | 0.060 |
| Sleep dissatisfaction | *Satisfied* | | | | | | |
| Slightly dissatisfied / very dissatisfied / unable to sleep | | 0.99 (0.85–1.14) | 0.839 | | | 0.95 (0.81–1.12) | 0.534 |
| Physical activity | *Every day* | | | | | | |
| 2–4 times a week / once a week / none | | 0.93 (0.84–1.03) | 0.156 | | | 0.92 (0.82–1.03) | 0.130 |
| Tsunami experience | *No* | | | | | | |
| Yes | | 0.99 (0.88–1.11) | 0.846 | | | 0.96 (0.85–1.09) | 0.561 |
| Nuclear accident experience | *No* | | | | | | |
| Yes | | 1.08 (0.98–1.19) | 0.135 | | | 1.03 (0.93–1.16) | 0.558 |
| Psychological distress | *K6 < 13* | | | | | | |
| K6 ≥ 13 | | 1.06 (0.91–1.25) | 0.446 | | | | |
| PTSD | *PCL < 44* | | | | | | |
| PCL ≥ 44 | | 1.02 (0.89–1.16) | 0.808 | | | 0.99 (0.85–1.15) | 0.903 |

Model 1: adjusted for age, evacuation status, BMI, systolic blood pressure, fasting blood glucose, triglycerides, eGFR, smoking status, drinking status, and unemployment experience. Model 2: adjusted for Model 1 plus sleep dissatisfaction, physical activity, tsunami experience, nuclear accident experience, and post-traumatic stress disorder. HR: hazard ratio, CI: confidence interval, SD: standard deviation, BMI: body mass index, HDL-C: high-density lipoprotein cholesterol, LDL-C: low-density lipoprotein cholesterol, eGFR: estimated glomerular filtration rate, K6: Kessler 6-item scale, PTSD: Post-traumatic stress disorder, PCL: post-traumatic stress disorder checklist.

We previously performed a cross-sectional study in which we reported the association between uric acid levels or hyperuricemia and psychological and social influences, such as evacuation due to disasters [21]. The previous study suggested that evacuation had significant positive associations with uric acid levels in both men and women. With hyperuricemia defined as uric acid levels > 7.0 mg/dL for men and > 6.0 mg/dL for women, significant associations were observed between evacuation and hyperuricemia in men, but not in women. The reason for the different results from the present study might be that the cross-sectional study did not have a time component. We believe that the present study, with the addition of a time

**Table 4. Cox regression analyses of the association between hyperuricemia (levels higher than 6 mg/dL) and earthquake-related items and lifestyle-related factors in 11,179 women.**

|  | Reference | Univariate | | Model 1 | | Model 2 | |
|---|---|---|---|---|---|---|---|
|  |  | HR (95% CI) | p | HR (95% CI) | p | HR (95% CI) | p |
| Age (years), *per 1 SD increase* |  | 1.19 (1.13–1.26) | <0.001 | 0.96 (0.88–1.05) | 0.356 | 0.97 (0.89–1.06) | 0.553 |
| Evacuation status | *Non-evacuee* |  |  |  |  |  |  |
| Evacuee |  | 1.22 (1.10–1.35) | <0.001 | 1.20 (1.06–1.34) | 0.003 | 1.18 (1.04–1.32) | 0.007 |
| BMI (kg/m$^2$), *per 1 SD increase* |  | 1.56 (1.49–1.63) | <0.001 | 1.49 (1.42–1.58) | <0.001 | 1.49 (1.41–1.58) | <0.001 |
| Systolic blood pressure (mmHg), *per 1 SD increase* |  | 1.36 (1.30–1.43) | <0.001 | 1.21 (1.14–1.28) | <0.001 | 1.21 (1.14–1.28) | <0.001 |
| Fasting blood glucose (mg/dL), *per 1 SD increase* |  | 1.15 (1.10–1.20) | <0.001 | 1.02 (0.97–1.07) | 0.479 | 1.02 (0.97–1.08) | 0.414 |
| HDL-C (mg/dL), *per 1 SD decrease* |  | 1.32 (1.25–1.40) | <0.001 |  |  |  |  |
| LDL-C (mg/dL), *per 1 SD increase* |  | 1.01 (0.96–1.07) | 0.659 |  |  |  |  |
| Triglycerides (mg/dL), *per 1 SD increase* |  | 1.32 (1.27–1.36) | <0.001 | 1.20 (1.15–1.26) | <0.001 | 1.20 (1.15–1.26) | <0.001 |
| eGFR (mL/min/1.73 m$^2$), *per 1 SD decrease* |  | 1.57 (1.48–1.66) | <0.001 | 1.57 (1.47–1.67) | <0.001 | 1.57 (1.47–1.67) | <0.001 |
| Smoking status | *Never smoker* |  |  |  |  |  |  |
| Quit smoking |  | 1.49 (1.22–1.82) | <0.001 | 1.41 (1.14–1.75) | 0.002 | 1.40 (1.13–1.74) | 0.002 |
| Current smoker |  | 1.46 (1.19–1.79) | <0.001 | 1.56 (1.23–1.97) | <0.001 | 1.56 (1.23–1.97) | <0.001 |
| Drinking status | *Never drinker* |  |  |  |  |  |  |
| Quit drinking |  | 2.09 (1.42–3.09) | <0.001 | 1.76 (1.17–2.64) | 0.007 | 1.73 (1.15–2.61) | 0.009 |
| <44 g/day |  | 1.25 (1.11–1.40) | <0.001 | 1.39 (1.23–1.58) | <0.001 | 1.39 (1.22–1.58) | <0.001 |
| ≥ 44 g/day |  | 2.67 (2.04–3.48) | <0.001 | 2.88 (2.11–3.92) | <0.001 | 2.88 (2.12–3.93) | <0.001 |
| Unemployment experience | *No* |  |  |  |  |  |  |
| Yes |  | 0.94 (0.82–1.07) | 0.348 | 0.92 (0.79–1.06) | 0.253 | 0.92 (0.79–1.06) | 0.235 |
| Sleep dissatisfaction | *Satisfied* |  |  |  |  |  |  |
| Slightly dissatisfied / very dissatisfied / unable to sleep |  | 1.10 (0.96–1.26) | 0.115 |  |  | 1.10 (0.95–1.28) | 0.208 |
| Physical activity | *Every day* |  |  |  |  |  |  |
| 2–4 times a week / once a week / none |  | 0.99 (0.89–1.11) | 0.888 |  |  | 1.07 (0.95–1.21) | 0.262 |
| Tsunami experience | *No* |  |  |  |  |  |  |
| Yes |  | 0.87 (0.75–1.01) | 0.060 |  |  | 0.87 (0.75–1.02) | 0.092 |
| Nuclear accident experience | *No* |  |  |  |  |  |  |
| Yes |  | 1.08 (0.97–1.20) | 0.182 |  |  | 1.06 (0.94–1.19) | 0.327 |
| Psychological distress | *K6 < 13* |  |  |  |  |  |  |
| K6 ≥ 13 |  | 1.09 (0.94–1.25) | 0.256 |  |  |  |  |
| PTSD | *PCL < 44* |  |  |  |  |  |  |
| PCL ≥ 44 |  | 1.12 (0.99–1.00) | 0.073 |  |  | 1.05 (0.92–1.21) | 0.469 |

Model 1: adjusted for age, evacuation status, BMI, systolic blood pressure, fasting blood glucose, triglycerides, eGFR, smoking status, drinking status and unemployment experience. Model 2: adjusted for Model 1 plus sleep dissatisfaction, physical activity, tsunami experience, nuclear accident experience, and post-traumatic stress disorder. HR: hazard ratio, CI: confidence interval, SD: standard deviation, BMI: body mass index, HDL-C: high-density lipoprotein cholesterol, LDL-C: low-density lipoprotein cholesterol, eGFR: estimated glomerular filtration rate, K6: Kessler 6-item scale, PTSD: Post-traumatic stress disorder, PCL: post-traumatic stress disorder checklist.

component, was better able to test whether evacuation following a natural disaster is associated with new-onset hyperuricemia under conditions with a higher level of evidence. In addition, Hashimoto *et al.* reported that evacuation after the earthquake associated with higher incidence of hyperuricemia using a pilot longitudinal data (n = 4,789) of 2-year in the FHMS [29]. The present study concluded the independent association of disaster evacuation to the new-onset of hyperuricemia adjusting for physical and socio-psychological factors using long-term longitudinal data of 7-year in the full cohort of FHMS (n = 18,140).

One possible reason for evacuation leading to hyperuricemia is a change in diet. Evacuation has been previously reported to exacerbate obesity by exacerbating lifestyle habits [16–20]. Since obesity is one of the risk factors for hyperuricemia [30], evacuation-associated obesity might led to the development of hyperuricemia. Furthermore, evacuees tended to consume more fructose-containing foods, such as juices [31]. A meta-analysis previously reported that high fructose intake is associated with hyperuricemia [32], and it remains possible that dietary changes with higher fructose intake following evacuation might be associated with the development of hyperuricemia, although it is quite unclear that this effect after the earthquake could also be obvious in the long term, as in the 7 year observation period in the present study. However, it is not conclusively known why disaster evacuation is one of the risk factors for new-onset hyperuricemia only in women and not in men. It is known that hyperinsulinemia causes increased uric acid reabsorption by urate transporter 1 (URAT1) and elevated serum uric acid levels. On the other hand, it has been reported that hyperglycemia and high urinary glucose levels decrease uric acid levels due to inhibition of uric acid reabsorption in the proximal tubules [33]. The impact and effect of evacuation on the development of hyperuricemia might, thus, vary depending on the presence or absence of diabetes at baseline. Prevalence rate of diabetes was higher in men than women (17.2% vs. 7.7%) at baseline in the present study. Therefore, the complication rate of diabetes and the frequency of diabetes development after the earthquake might be one of the reasons for the differences between men and women. Hence, the detailed mechanisms of hyperuricemia among evacuees after a natural disaster and the reasons for the sex-related difference need to be examined in the future.

There might be objections to the clinical implications of this study, because it uses a less stringent definition of hyperuricemia for women than for men. However, it has been suggested that uric acid levels are less likely to increase in women than in men. This difference is said to involve the promotion of uric acid excretion by estrogen [34], and the cutoff value of uric acid levels between men and women has been discussed in previous literature [23–25]. One study reported that hyperuricemia is an independent risk factor for end-stage renal disease (ESRD) in women but not in men, when hyperuricemia is defined as $\geq 7$ and $\geq 6$ mg/dL in men and women, respectively [35]. Therefore, we believe that it is also clinically appropriate that the definition of hyperuricemia differs between men and women in this study. However, differences in the risk factors for hyperuricemia between men and women need to be examined in greater detail in future.

The present study showed that evacuation after a natural disaster is independently associated with the development of new-onset hyperuricemia. It has been previously reported that evacuation after a natural disaster is an independent risk factor for metabolic syndrome, diabetes mellitus, dyslipidemia, hypertension, and chronic kidney disease [16–20, 36]. These are all risk factors for cardiovascular events and end-stage renal failure, and hence, are important issues that directly affect the health of disaster victims and should be observed for in the long term in victims of natural disasters requiring evacuation.

The strength of the present study was a large-scale observational study which showed significant association between evacuation experience after the earthquake and new-onset of hyperuricemia in the long-term of 7-year observation. This means that the impact of evacuation on hyperuricemia is not necessarily a temporary problem, but should be viewed as a longer-term and future health care issue. Also, interestingly, evacuation was shown to be a significant risk factor for the new-onset of hyperuricemia in women aged < 65 years, with BMI < 25.0 kg/m$^2$, never smokers, those not on anti-hypertensive medication, and with no history of diabetes, or abnormal renal function (S4 Table). We believe that this information is important from a risk management perspective, since apparently healthy young women without lifestyle-related diseases might newly develop hyperuricemia as a result of evacuation. Our study suggests that after

a natural disaster, it is very important to monitor the health status of the victims, including for the presence of hyperuricemia, and it is necessary to return their living environment to normal as soon as possible, and to improve their lifestyle, including diet, drinking, and exercise habits.

There are several limitations to the present study. First, the adjustment factors for hyperuricemia including lifestyle characteristics were only assessed and measured at baseline. We could not assess how these adjustment factors changed over the observation period in the present study, therefore, we only studied whether these factors at baseline affected the risk of developing hyperuricemia, although the changes in these factors might affect the results. Second, as the present study consisted of residents living near the Fukushima Daiichi Nuclear Power Plant in Fukushima prefecture who completed the Comprehensive Health Check of the FHMS, the residents who did not completed the Comprehensive Health Check were excluded from the analysis. Although the reasons why they did not complete the health check were unclear, the exclusion of them could be a selection bias and might affect the results of the present study. Third, the detailed mechanism by which disaster evacuation causes hyperuricemia remains unclear. Evacuation itself may not be the direct cause of hyperuricemia. Factors related to evacuation, such as severe disaster damage and changes in living environment and lifestyle may be associated with the increased risk of new onset of hyperuricemia, but lifestyle characteristics were measured only at baseline. Therefore, future research is still needed to investigate with longitudinal evaluation of clinical, biochemical, and lifestyle characteristics whether exacerbating lifestyle habits after a natural disaster are caused by changes in the living environment associated with relocation, by the stress associated with the disaster damage, or by the other factors, although exacerbating lifestyle habits such as changes in diet are thought to be one of the causes of hyperuricemia. Forth, data on the duration of evacuation were not available in the present study, although long-term evacuation could have significant effects on health damage of victims. Fifth, we did not assess medications (with or without treatment) for hyperuricemia. Pharmacotherapy can significantly reduce uric acid levels and thus have a significant impact on the results of the present study. We believe that the impact of drug therapy is not small, as there are several types and capacities of drugs that directly lower uric acid levels, as well as several drugs that indirectly lower uric acid levels. Finally, we did not assess diet. Previous reports suggest that uric acid levels are greatly influenced by dietary content [32].

In conclusion, several lifestyle- and disaster-related factors, particularly evacuation, were found to be risk factors associated with new-onset hyperuricemia after the Great East Japan Earthquake and the Fukushima Daiichi Nuclear Power Plant accident. We should consider the possibility that hyperuricemia might develop in response to natural disasters. Further studies are needed to determine whether careful health management for disaster victims or improvements in the living environment, such as the quality of food provided and sleep, in evacuation centers, reduce the risk of hyperuricemia and health damage such as disaster-related death in disaster victims.

## Supporting information

**S1 Table. Clinical and biochemical characteristics of the 18,140 participants stratified by sex and the development of hyperuricemia.** The values in the table indicate the average value (standard deviation) or the percentage. Fasting blood glucose, triglycerides, AST, ALT, and γ-GT are reported as the median (25–75% percentile). BMI: body mass index, HbA1c: hemoglobin A1c, HDL-C: high-density lipoprotein cholesterol, LDL-C: low-density lipoprotein cholesterol, AST: aspartate aminotransferase, ALT: alanine aminotransferase, γ-GT: γ-glutamyl transpeptidase, eGFR: estimated glomerular filtration rate.
(DOCX)

**S2 Table. Lifestyle characteristics of the 18,140 participants stratified by sex and the development of hyperuricemia.** The values in the table indicate the number (percentage). K6: Kessler 6-item scale, PTSD: Post-traumatic stress disorder, PCL: Post-traumatic Stress Disorder Checklist.
(DOCX)

**S3 Table. Hazard ratios of evacuation for the development of hyperuricemia (levels higher than 7 mg/dL) in 6,961 men, according to baseline variates.** Model 1: adjusted for age, BMI, systolic blood pressure, fasting blood glucose, triglycerides, eGFR, smoking status, drinking status, and unemployment experience. Model 2: adjusted for Model 1 plus sleep dissatisfaction, physical activity, tsunami experience, nuclear accident experience and, post-traumatic stress disorder. HR: hazard ratio, CI: confidence interval, BMI: body mass index.
(DOCX)

**S4 Table. Hazard ratio of evacuation for the development of hyperuricemia (levels higher than 6 mg/dL) in 11,179 women, according to baseline variates.** Model 1: adjusted for age, BMI, systolic blood pressure, fasting blood glucose, triglycerides, eGFR, smoking status, drinking status, and unemployment experience. Model 2: adjusted for Model 1 plus sleep dissatisfaction, physical activity, tsunami experience, nuclear accident experience and, post-traumatic stress disorder. HR: hazard ratio, CI: confidence interval, BMI: body mass index.
(DOCX)

## Acknowledgments

We thank the expert committee members, advisors, and staff of the Fukushima Health Survey Group for conducting this survey and for their support. The findings and conclusions of this article are solely the responsibility of the authors and do not represent the official views of the Fukushima Prefectural Government.

## Author Contributions

**Conceptualization:** Kenichi Tanaka, Tetsuya Ohira.

**Formal analysis:** Kanako Okazaki, Hironori Nakano, Masanori Nagao.

**Methodology:** Kenichi Tanaka, Tetsuya Ohira.

**Project administration:** Tetsuya Ohira, Akira Sakai, Seiji Yasumura, Masaharu Maeda, Hirooki Yabe, Mitsuaki Hosoya, Atsushi Takahashi, Mayumi Harigane, Fumikazu Hayashi, Michio Shimabukuro, Hitoshi Ohto, Kenji Kamiya, Junichiro J. Kazama.

**Supervision:** Kenji Kamiya, Junichiro J. Kazama.

**Writing – original draft:** Kazuya Honda, Kenichi Tanaka.

**Writing – review & editing:** Kenichi Tanaka, Eri Kobari, Sakumi Kazama, Shigeatsu Hashimoto, Tetsuya Ohira, Akira Sakai, Seiji Yasumura, Masaharu Maeda, Hirooki Yabe, Mitsuaki Hosoya, Atsushi Takahashi, Mayumi Harigane, Fumikazu Hayashi, Michio Shimabukuro, Hitoshi Ohto, Kenji Kamiya, Junichiro J. Kazama.

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
