## [Decision Letter · Decision Letter 0]

12 Jun 2023

PONE-D-23-01408Relationship between evacuation after the Great East Japan Earthquake and new-onset hyperuricemia: A 7-year prospective longitudinal study of the Fukushima Health Management SurveyPLOS ONE

Dear Dr. Tanaka,

Thank you for submitting your manuscript to PLOS ONE. After careful consideration, we feel that it has merit but does not fully meet PLOS ONE’s publication criteria as it currently stands. Therefore, we invite you to submit a revised version of the manuscript that addresses the points raised during the review process.

Additional Editor Comments:

According to my review of the manuscript, the following issues were considered. The authors/corresponding author should precisely respond to issues 1-4.

1. A discrepancy was found in the participants included in the two consort diagrams of published article and manuscript.

2. All the materials and methods sections (psychological distress definition, participants' classification score, demographic, clinical, and biochemical measurements) were the same, except for applying the Cox proportional hazards regression model that authors added to analyze data of their work.

3. The grant number of both studies is the same. However, there are many similarities between the two articles and the manuscript. Undoubtedly, it is part of a salami slicing in publication.

4. The authors did not mention anything about evacuees and non-evacuees in the published article despite describing its terms. However, they mentioned evacuation and non-evacuation in the published article; and evacuees and non-evacuees in the manuscript. 

We look forward to receiving your revised manuscript.

Kind regards,

Mohammad Reza Mahmoodi, Ph.D.

Academic Editor

PLOS ONE 

Journal Requirements:

"This work was supported by the National Health Fund for Children and Adults Affected by the Nuclear Incident."

"This work was supported by the National Health Fund for Children and Adults Affected by the Nuclear Incident."

Reviewers' comments:

Reviewer's Responses to Questions

**Comments to the Author**

1. Is the manuscript technically sound, and do the data support the conclusions?

Reviewer #1: Yes

Reviewer #2: Yes

Reviewer #3: Yes

Reviewer #4: Partly

2. Has the statistical analysis been performed appropriately and rigorously? 

Reviewer #1: Yes

Reviewer #2: Yes

Reviewer #3: I Don't Know

Reviewer #4: Yes

3. Have the authors made all data underlying the findings in their manuscript fully available?

Reviewer #1: Yes

Reviewer #2: Yes

Reviewer #3: Yes

Reviewer #4: Yes

4. Is the manuscript presented in an intelligible fashion and written in standard English?

Reviewer #1: Yes

Reviewer #2: Yes

Reviewer #3: Yes

Reviewer #4: Yes

5. Review Comments to the Author

Reviewer #1: This is an interesting paper and the content is very good but I suggest the paper can be improved in the following ways:

Abstract

-Please modify the headlines based on the format of the journal

-In the conclusion part, it is necessary to specify the researcher's proposal to improve the conditions and use of the beneficiaries.

Introduction

Please bring the following items

1- Definition of the research problem

2- The magnitude and importance of the study variable

3- Bringing the variable epidemiology information of the study at the world level and the place of study

4- Bringing theoretical knowledge about the importance of studying

5- Expressing the necessity of conducting the study

Finally, the practical purpose of the study should be stated

Methods

-please declare scoring of the used tool in this study.

- What steps have you taken in your study to ensure reliability of study tool in your study participants?

Discussion

In the discussion section, it is necessary to compare the main results of the study with the results of other studies in this field

What are the strengths and limitations of the study?

Conclusion

What are your suggestion for future studies?

Best regard

Reviewer #2: 1- In the statistical analysis section and in line 216 the authors said they performed two separate Cox models for men and women. Please clarify why you use two separate models. Does the PH assumption not satisfy for sex variable? please perform a model with the sex variable included and report the PH assumption for the sex variable.

2- In line 227, please clarify the missing imputation method.

3- In page 11, why the authors report mean(SD) for some continuous variables and median(CI) for some other continuous variables? please use the same summary statistics for continuous variables.

4- In line 287, please replace the "multivariate" with "multivariable". Because a multivariate model is used when you have more than one response variable.

5- No clear explanations in line 287. Explain why the authors used two models ( model 1 and model 2 in table 3). please clarify in detail.

6- What does "Every time +1 SD" means in table 3? Is it the guideline of the journal? if not, please keep blank the reference level for continuous variables.

7- In discussion, the authors referred to their cross sectional study. Please clarify why there is a different in 33559 in this study and 33493 in cross sectional study?

8- Please add the consort diagram in this study like the cross-sectional study.

Reviewer #3: Thank you for offering the opportunity to review this paper. I find the paper well written. The most important strength of this manusdript is the complete explanation of the method. I only have some small minor comments.

Please find below the detailed list of my comments. Best regards

1-the caption of Table 2 needs to be improve because the items related to the disaster are not mentioned in the caption while they are in the table.

2-Although the sampling method is explained in detail, the name of the sampling method is not mentioned

3- Because the sampling method is not random, the results cannot be generalized and this is one of the limitations of the study

Reviewer #4: This is an interesting manuscript that has many strengths including some important findings. However, the manuscript could benefit from a revision that addresses several issues. First, more information is needed about exactly when (and how many times) assessments were done of the key variables in the study. The authors mention that baseline assessments were done in 2011 and that follow-up Comprehensive Health Check assessments were done in 2012-17. It appears, however, that all of the analyses use only the baseline data as predictors of developing new onset hyperuicemia and that there is no longitudinal analysis of how these other predictor variables may change over time. This should be clarified. Second, the key variable of evacuation as used in this study is technically accurate but somewhat misleading. It appears that evacuation was not voluntary but mandatory based on specific communities that had either experienced a great deal of damage due to the earthquake or tsunami or that were viewed as being at high risk of radiation exposure. In other words, evacuation not only means that you had to relocate but is also a proxy for where you lived before the disaster and how badly where you lived was affected by the disaster. Relocation can clearly affect important aspects of your lifestyle such as diet, physical activity, access to usual healthcare, etc., but if relocation is mandatory and prompted by a high level of disaster exposure, stress related to high levels of disaster exposure whether or not you had to evacuate also comes into play. This should be acknowledged. Third, there is no mention of the duration of the evacuation or inclusion of a measure of this in the analyses. In many natural disasters, evacuations are not permanent, and many individuals return to the disaster sites within days, weeks, or months. Trying to rebuild your property and life after a disaster is stressful whether you have returned to the site were not, but providing additional information about the length of the evacuation is important under either circumstance. Fourth, if the lifestyle characteristic measures listed in Table 2 (including past 30 day psychological distress and PTSD) were measured only at baseline, there would be no way to determine whether these variables changed over time or whether any such changes are related to changes in uric acid levels. This would be a substantial limitation that should be acknowledged. Fifth, it is unclear whether the key uric acid level measure in the study took into account individuals who were receiving medication to reduce high uric acid levels. Such medications typically reduce uric acid levels below the threshold for hyperuiecemia used in the study, so this is relevant information. Sixth, as reflected in Figure 1, there was substantial nonparticipation and attrition in the study, raising the possibility of bias in study findings. It would be useful if the authors could address this issue. Finally, the overall finding that individuals who were forced to be evacuated due to major disaster damage or threat of radiation exposure where they lived had higher risk of developing excessive uric acid levels is important. The difficulty is that evacuation per se is not likely to be the primary mechanism accounting for this change. Therefore, evacuation is a proxy for high disaster exposure, so the question remains how does such exposure relate to known biopsychosocial mechanisms that might account for increases in uric acid levels. The manuscript would benefit from a discussion that acknowledges more clearly that the study does not have the data to test some of the potential mechanisms, as I do not think that most of the predictor variables measured at baseline were measured longitudinally.

6. PLOS authors have the option to publish the peer review history of their article (what does this mean?). If published, this will include your full peer review and any attached files.

Reviewer #1: No

Reviewer #2: No

Reviewer #3: No

Reviewer #4: No

---

## [Author Response · Author response to Decision Letter 0]

12 Sep 2023

Thank you for your constructive and excellent criticism and suggestions. We hope that we have responded adequately. 

The changes made to the revised manuscript are shown by yellow highlights.

Additional Editor Comments:

According to my review of the manuscript, the following issues were considered. The authors/corresponding author should precisely respond to issues 1-4.

1. A discrepancy was found in the participants included in the two consort diagrams of published article and manuscript.　

Respond: In our published cross-sectional study, the age of the subjects was extracted by age at the time of medical examination. In the present longitudinal study, the subjects were selected by age at the beginning of fiscal year (April 1, 2011). So, the number of subjects slightly differed because of the different extraction methods, although the target age groups were the same.

2. All the materials and methods sections (psychological distress definition, participants' classification score, demographic, clinical, and biochemical measurements) were the same, except for applying the Cox proportional hazards regression model that authors added to analyze data of their work. 

3. The grant number of both studies is the same. However, there are many similarities between the two articles and the manuscript. Undoubtedly, it is part of a salami slicing in publication. 

Respond (for 2 and 3): Thank you for your comments on this point. We have reported a study that revealed the relationship between evacuation after the earthquake and the prevalence of hyperuricemia (Honda K, et al. Evacuation after the Great East Japan Earthquake is an independent factor associated with hyperuricemia: The Fukushima Health Management Survey. Nutr Metab Cardiovasc Dis. 2021;31(4):1177-88.). As the editor pointed out, psychological distress definition, participants' classification score, demographic, clinical, and biochemical measurements in the method section were same as the present study, however, the published article only included a cross-sectional analysis using a baseline data, and revealed evacuation had significant and positive association with the prevalence of hyperuricemia. In the present prospective study, subject with hyperuricemia at baseline were excluded, and those who did not have hyperuricemia as of fiscal year 2011 were followed for long-term (7 years) to investigate whether evacuation related to increased risk for new-onset of hyperuricemia adjusting for physical and socio-psychological factors. The present study showed evacuation after a natural disaster was a significant risk factor for the new-onset of hyperuricemia in women independent from physical and socio-psychological factors. Therefore, the present study includes a novel finding different from our published article and the authors consider the present study well worth publishing in the journal.

4. The authors did not mention anything about evacuees and non-evacuees in the published article despite describing its terms. However, they mentioned evacuation and non-evacuation in the published article; and evacuees and non-evacuees in the manuscript. 

Respond: As suggested by the editor, we revised related descriptions about the definitions of evacuation, evacuee, and non-evacuee in the methods section (line 212 to 219).

I hope that you will find all of the changes we made satisfactory. We would like to thank you for your suggestions that were of invaluable help in considerably improving the quality of the manuscript.

Thank you in advance for your attention.

Looking forward to hearing from you.

Best regards

Kenichi Tanaka, MD, PhD

Journal Requirements:

Responds: We corrected the display in the manuscript in accordance with PLOS ONE's style requirements.

Respond: We revised the related descriptions in the method section (line 155 to 158).

"This work was supported by the National Health Fund for Children and Adults Affected by the Nuclear Incident."

"This work was supported by the National Health Fund for Children and Adults Affected by the Nuclear Incident."

Respond (for 3 and 4): We revised and added the related descriptions in the manuscript (line 491 to 496) and the cover letter. Revised descriptions were as below.

“Funding: This work was supported by the National Health Fund for Children and Adults Affected by the Nuclear Incident. No financial support has been received from any source other than this fund. There was no additional external funding received for this study. The funders had no role in study design, data collection and analysis, decision to publish, or preparation of the manuscript.”

Respond: We added the related descriptions in the cover letter. Added descriptions were as below.

“Data Availability Statement: Data underlying the findings in this study cannot be made publicly available due the nature of ethical approval for the study. Interested researchers may submit requests to the Fukushima Medical University’s Ethics Committee (Contact information: Email: rs@fmu.ac.jp) for access to confidential data.”

 

Reviewers' comments:

Reviewer's Responses to Questions

Comments to the Author

1. Is the manuscript technically sound, and do the data support the conclusions?

Reviewer #1: Yes

Reviewer #2: Yes

Reviewer #3: Yes

Reviewer #4: Partly

2. Has the statistical analysis been performed appropriately and rigorously? 

Reviewer #1: Yes

Reviewer #2: Yes

Reviewer #3: I Don't Know

Reviewer #4: Yes

3. Have the authors made all data underlying the findings in their manuscript fully available?

Reviewer #1: Yes

Reviewer #2: Yes

Reviewer #3: Yes

Reviewer #4: Yes

4. Is the manuscript presented in an intelligible fashion and written in standard English?

Reviewer #1: Yes

Reviewer #2: Yes

Reviewer #3: Yes

Reviewer #4: Yes

5. Review Comments to the Author

 

Thank you for your constructive and excellent criticism and suggestions. We hope that we have responded adequately. 

The changes made to the revised manuscript are shown by yellow highlights.

Reviewer #1: This is an interesting paper and the content is very good but I suggest the paper can be improved in the following ways:

Abstract

-Please modify the headlines based on the format of the journal

Respond: As suggested by the reviewer, we modified the headlines in the abstract section based on the format of the journal.

-In the conclusion part, it is necessary to specify the researcher's proposal to improve the conditions and use of the beneficiaries.

Respond: As suggested by the reviewer, we added related descriptions in discussion in the abstract section (line 83 to 84).

Introduction

Please bring the following items

1- Definition of the research problem

2- The magnitude and importance of the study variable

3- Bringing the variable epidemiology information of the study at the world level and the place of study

4- Bringing theoretical knowledge about the importance of studying

5- Expressing the necessity of conducting the study

Finally, the practical purpose of the study should be stated

Respond: As suggested by the reviewer, we revised and added the related descriptions in the introduction section (line 105 to 115).

Methods

-please declare scoring of the used tool in this study.

- What steps have you taken in your study to ensure reliability of study tool in your study participants?

Respond: Thank you for your comments in these points. In the Mental Health and Lifestyle Survey, the Japanese versions of the Kessler 6-item scale (K6) and Post-traumatic Stress Disorder Checklist (PCL) were used to assess the participants’ mental health. The K6 consists of six brief questions about depressive and anxiety symptoms during the past 30 days, with overall scores ranging from 0 to 24. We defined psychological distress as corresponding to a K6 score ≥ 13. The PCL is a tool used to evaluate symptoms of post-traumatic stress disorder (PTSD) during the past 30 days. The PCL consists of 17 items, and the overall score ranges from 17 to 85. We classified participants as having probable PTSD if their overall PCL score was ≥ 44. Detailed information and their reliability were described in previous studies ([15] Suzuki Y, et al. Diagnostic accuracy of Japanese posttraumatic stress measures after a complex disaster: The Fukushima Health Management Survey. Asia Pac Psychiatry. 2017;9(1). [26] Furukawa TA, et al. The performance of the Japanese version of the K6 and K10 in the World Mental Health Survey Japan. Int J Methods Psychiatr Res. 2008;17(3):152-8. [27] Kessler RC, et al. Screening for serious mental illness in the general population. Arch Gen Psychiatry. 2003;60(2):184-9.). Related and added descriptions were in the methods section (line 194 to 203).

Discussion

In the discussion section, it is necessary to compare the main results of the study with the results of other studies in this field

Respond: As suggested by the reviewer, we added related descriptions in the discussion section (line 354 to 359). 

What are the strengths and limitations of the study? 

Respond: As suggested by the reviewer and the other reviewers, we added and revised descriptions of limitations of the study (line 418 to 445). Also, the strengths of the study were described in line 403 to 417.

Conclusion

What are your suggestion for future studies? 

Respond: Thank you for your comments on these points. We consider that improvements in the living environment, such as the quality of food provided and sleep, in evacuation centers, reduce the risk of hyperuricemia and health damage such as disaster-related death in disaster victims. We added related descriptions in the discussion section (line 450 to 455).

Minor error was found in number of municipalities, so, we corrected its number from nine to thirteen (line 213).

I hope that you will find all of the changes we made satisfactory. We would like to thank you for your suggestions that were of invaluable help in considerably improving the quality of the manuscript.

Thank you in advance for your attention.

Looking forward to hearing from you.

Best regards

Kenichi Tanaka, MD, PhD

 

Thank you for your constructive and excellent criticism and suggestions. We hope that we have responded adequately. 

The changes made to the revised manuscript are shown by yellow highlights.

Reviewer #2: 1- In the statistical analysis section and in line 216 the authors said they performed two separate Cox models for men and women. Please clarify why you use two separate models. Does the PH assumption not satisfy for sex variable? please perform a model with the sex variable included and report the PH assumption for the sex variable. 

Respond: As the cutoff values for hyperuricemia were different between men and women and the clinical impact of hyperuricemia might also be different, Cox proportional hazards regression model analyses with the onset of hyperuricemia as the dependent variable were performed separately for men and women. Indeed, a significant difference was observed in the effect on evacuation for the risk of hyperuricemia between men and women (interaction P<0.001). We added related descriptions in statistical analysis in the method section (line 225 to 228).

2- In line 227, please clarify the missing imputation method.

Respond: In the present study, the dependent variable excluded missing data, while the explanatory variables included missing data. Specifically, dummy variables were created for missing and non-missing values, with missing values set to "1" and non-missing values set to "0" and used in the multivariate analysis. We revised related descriptions in statistical analysis in the method section (line 239 to 242).

3- In page 11, why the authors report mean(SD) for some continuous variables and median(CI) for some other continuous variables? please use the same summary statistics for continuous variables. 

Respond: Thank you for your comments on this point.　In the present study, the means (SD) were used for continuous variables for which normal distributions were confirmed, and the medians (25-75%) were used for which normal distributions could not be confirmed. We revised related descriptions in Table 1.

4- In line 287, please replace the "multivariate" with "multivariable". Because a multivariate model is used when you have more than one response variable.

Respond: As suggested by the editor, we replaced the "multivariate" with "multivariable" (line 300).

5- No clear explanations in line 287. Explain why the authors used two models ( model 1 and model 2 in table 3). please clarify in detail. 

Respond: We added and revised related description in statistical analysis in the method section (line 228 to 238). Related descriptions were as below.

“First, referring to previous cross-sectional studies [21], we adjusted for the independent variables of age, evacuation, BMI, systolic blood pressure, fasting blood glucose, triglycerides, eGFR, smoking status, drinking status, and unemployment in Model 1. Next, in order to consider physical and socio-psychological factors, in addition to Model 1, sleep dissatisfaction, physical activity, tsunami experience, nuclear accident experience, and PTSD were adjusted as independent variables for Model 2. Since HDL-C, LDL-C, and triglycerides statistically strongly correlate with each other, triglyceride level was adopted as representative of dyslipidemia, as in previous reports [28]. Since there was a strong correlation between K6 score and PCL score, PTSD (PCL score ≥ 44) was adopted as an independent factor in this study.”

6- What does "Every time +1 SD" means in table 3? Is it the guideline of the journal? if not, please keep blank the reference level for continuous variables. 

Respond: Thank you for your comments on this point. "Every time +1 SD" means hazard ratio per 1 SD increase of these continuous variables. These displays may be little difficult to understand, so we revised them in Table 3 and 4. Also, hazard ratios of age were not analyzed as per 1 SD increase but as per 10 years increase. So, Cox regression analyses were re-analyzed with “age per 1 SD increase” and made minor revisions in data in Table 3 and 4.

7- In discussion, the authors referred to their cross sectional study. Please clarify why there is a different in 33559 in this study and 33493 in cross sectional study? 

Respond: In our published cross-sectional study, the age of the subjects was extracted by age at the time of medical examination. In the present longitudinal study, the subjects were selected by age at the beginning of the fiscal year (April 1, 2011). So, the number of subjects slightly differed because of the different extraction methods, although the target age groups were the same.

8- Please add the consort diagram in this study like the cross-sectional study. 

Respond: The consort diagram was displayed in figure 1. Please see figure 1.

Minor error was found in number of municipalities, so, we corrected its number from nine to thirteen (line 213).

I hope that you will find all of the changes we made satisfactory. We would like to thank you for your suggestions that were of invaluable help in considerably improving the quality of the manuscript.

Thank you in advance for your attention.

Looking forward to hearing from you.

Best regards

Kenichi Tanaka, MD, PhD

 

Thank you for your constructive and excellent criticism and suggestions. We hope that we have responded adequately. 

The changes made to the revised manuscript are shown by yellow highlights.

Reviewer #3: Thank you for offering the opportunity to review this paper. I find the paper well written. The most important strength of this manusdript is the complete explanation of the method. I only have some small minor comments.

Please find below the detailed list of my comments. Best regards

1-the caption of Table 2 needs to be improve because the items related to the disaster are not mentioned in the caption while they are in the table. 

Respond: As suggested by the reviewer, we improved the caption of Table 2.

2-Although the sampling method is explained in detail, the name of the sampling method is not mentioned Respond: Thank you for your comments on this point. The present study was a cohort study included all residents between 40 and 74 years living in 13 communities near the Fukushima Daiichi Nuclear Power Plant in Fukushima prefecture who completed the Comprehensive Health Check of the FHMS in fiscal year 2011, so any sampling method was not used in the present study.

3- Because the sampling method is not random, the results cannot be generalized and this is one of the limitations of the study 

Respond: As suggested by the reviewer, we added related descriptions as one of the study limitations (line 423 to 428). Added descriptions were as below.

“Second, as the present study consisted of residents living near the Fukushima Daiichi Nuclear Power Plant in Fukushima prefecture who completed the Comprehensive Health Check of the FHMS, the residents who did not completed the Comprehensive Health Check were excluded from the analysis. Although the reasons why they did not complete the health check were unclear, the exclusion of them could be a selection bias and might affect the results of the present study.”

Minor error was found in number of municipalities, so, we corrected its number from nine to thirteen (line 213).

I hope that you will find all of the changes we made satisfactory. We would like to thank you for your suggestions that were of invaluable help in considerably improving the quality of the manuscript.

Thank you in advance for your attention.

Looking forward to hearing from you.

Best regards

Kenichi Tanaka, MD, PhD

 

Thank you for your constructive and excellent criticism and suggestions. We hope that we have responded adequately. 

The changes made to the revised manuscript are shown by yellow highlights.

Reviewer #4: This is an interesting manuscript that has many strengths including some important findings. However, the manuscript could benefit from a revision that addresses several issues. 

First, more information is needed about exactly when (and how many times) assessments were done of the key variables in the study. The authors mention that baseline assessments were done in 2011 and that follow-up Comprehensive Health Check assessments were done in 2012-17. It appears, however, that all of the analyses use only the baseline data as predictors of developing new onset hyperuicemia and that there is no longitudinal analysis of how these other predictor variables may change over time. This should be clarified. 

Respond: Your comment on this point is extremely relevant.　We could not assess longitudinal changes in baseline variables. So, we added and revised descriptions as a study limitation in the discussion section (line 418 to 423). Added descriptions were as below.

“First, the adjustment factors for hyperuricemia including lifestyle characteristics were only assessed and measured at baseline. We could not assess how these adjustment factors changed over the observation period in the present study, therefore, we only studied whether these factors at baseline affected the risk of developing hyperuricemia, although the changes in these factors might affect the results.”

Second, the key variable of evacuation as used in this study is technically accurate but somewhat misleading. It appears that evacuation was not voluntary but mandatory based on specific communities that had either experienced a great deal of damage due to the earthquake or tsunami or that were viewed as being at high risk of radiation exposure. In other words, evacuation not only means that you had to relocate but is also a proxy for where you lived before the disaster and how badly where you lived was affected by the disaster. Relocation can clearly affect important aspects of your lifestyle such as diet, physical activity, access to usual healthcare, etc., but if relocation is mandatory and prompted by a high level of disaster exposure, stress related to high levels of disaster exposure whether or not you had to evacuate also comes into play. This should be acknowledged. 

Respond: Thank you for your comments on this point. We considered this point to be extremely important when taking measures after a natural disaster. So, we added related descriptions as a study limitation in the discussion section (line 428 to 439). Added descriptions were as below.

“Third, the detailed mechanism by which disaster evacuation causes hyperuricemia remains unclear. Evacuation itself may not be the direct cause of hyperuricemia. Factors related to evacuation, such as severe disaster damage and changes in living environment and lifestyle may be associated with the increased risk of new onset of hyperuricemia, but lifestyle characteristics were measured only at baseline. Therefore, future research is still needed to investigate with longitudinal evaluation of clinical, biochemical, and lifestyle characteristics whether exacerbating lifestyle habits after a natural disaster are caused by changes in the living environment associated with relocation, by the stress associated with the disaster damage, or by the other factors, although exacerbating lifestyle habits such as changes in diet are thought to be one of the causes of hyperuricemia.”

Third, there is no mention of the duration of the evacuation or inclusion of a measure of this in the analyses. In many natural disasters, evacuations are not permanent, and many individuals return to the disaster sites within days, weeks, or months. Trying to rebuild your property and life after a disaster is stressful whether you have returned to the site were not, but providing additional information about the length of the evacuation is important under either circumstance. 

Respond: Thank you for your comments on this point. We added related descriptions as a study limitation in the discussion section (line 439 to 441). Added descriptions were as below.

“Forth, data on the duration of evacuation were not available in the present study, although long-term evacuation could have significant effects on health damage of victims.”

Fourth, if the lifestyle characteristic measures listed in Table 2 (including past 30 day psychological distress and PTSD) were measured only at baseline, there would be no way to determine whether these variables changed over time or whether any such changes are related to changes in uric acid levels. This would be a substantial limitation that should be acknowledged. 

Respond: As the reviewer pointed out, we could not assess longitudinal changes in baseline variables. So, we added and revised descriptions as a study limitation in the discussion section (line 418 to 423). Added descriptions were as below.

“First, the adjustment factors for hyperuricemia including lifestyle characteristics were only assessed and measured at baseline. We could not assess how these adjustment factors changed over the observation period in the present study, therefore, we only studied whether these factors at baseline affected the risk of developing hyperuricemia, although the changes in these factors might affect the results.”

Fifth, it is unclear whether the key uric acid level measure in the study took into account individuals who were receiving medication to reduce high uric acid levels. Such medications typically reduce uric acid levels below the threshold for hyperuiecemia used in the study, so this is relevant information. 

Respond: Your comment on this point is extremely relevant.　We could not assess data on medications for hyperuricemia. So, we added and revised descriptions as a study limitation in the discussion section (line 441 to 445). Added descriptions were as below.

“Fifth, we did not assess medications (with or without treatment) for hyperuricemia. Pharmacotherapy can significantly reduce uric acid levels and thus have a significant impact on the results of the present study. We believe that the impact of drug therapy is not small, as there are several types and capacities of drugs that directly lower uric acid levels, as well as several drugs that indirectly lower uric acid levels.”

Sixth, as reflected in Figure 1, there was substantial nonparticipation and attrition in the study, raising the possibility of bias in study findings. It would be useful if the authors could address this issue. 

Respond: As suggested by the reviewer, we added related descriptions as one of the study limitations (line 423 to 428). Added descriptions were as below.

“Second, as the present study consisted of residents living near the Fukushima Daiichi Nuclear Power Plant in Fukushima prefecture who completed the Comprehensive Health Check of the FHMS, the residents who did not completed the Comprehensive Health Check were excluded from the analysis. Although the reasons why they did not complete the health check were unclear, the exclusion of them could be a selection bias and might affect the results of the present study.”

Finally, the overall finding that individuals who were forced to be evacuated due to major disaster damage or threat of radiation exposure where they lived had higher risk of developing excessive uric acid levels is important. The difficulty is that evacuation per se is not likely to be the primary mechanism accounting for this change. Therefore, evacuation is a proxy for high disaster exposure, so the question remains how does such exposure relate to known biopsychosocial mechanisms that might account for increases in uric acid levels. The manuscript would benefit from a discussion that acknowledges more clearly that the study does not have the data to test some of the potential mechanisms, as I do not think that most of the predictor variables measured at baseline were measured longitudinally. 

Respond: Thank you for your extremely valuable comments on this point. We added related descriptions as one of the study limitations (line 428 to 439). Added descriptions were as below. These descriptions were related to second comment by the reviewer.

 “Third, the detailed mechanism by which disaster evacuation causes hyperuricemia remains unclear. Evacuation itself may not be the direct cause of hyperuricemia. Factors related to evacuation, such as severe disaster damage and changes in living environment and lifestyle may be associated with the increased risk of new onset of hyperuricemia, but lifestyle characteristics were measured only at baseline. Therefore, future research is still needed to investigate with longitudinal evaluation of clinical, biochemical, and lifestyle characteristics whether exacerbating lifestyle habits after a natural disaster are caused by changes in the living environment associated with relocation, by the stress associated with the disaster damage, or by the other factors, although exacerbating lifestyle habits such as changes in diet are thought to be one of the causes of hyperuricemia.”

Minor error was found in number of municipalities, so, we corrected its number from nine to thirteen (line 213).

I hope that you will find all of the changes we made satisfactory. We would like to thank you for your suggestions that were of invaluable help in considerably improving the quality of the manuscript.

Thank you in advance for your attention.

Looking forward to hearing from you.

Best regards

Kenichi Tanaka, MD, PhD

---

## [Decision Letter · Decision Letter 1]

13 Oct 2023

Relationship between evacuation after the Great East Japan Earthquake and new-onset hyperuricemia: A 7-year prospective longitudinal study of the Fukushima Health Management Survey

PONE-D-23-01408R1

Dear Dr. Tanaka,

We’re pleased to inform you that your manuscript has been judged scientifically suitable for publication and will be formally accepted for publication once it meets all outstanding technical requirements.

Kind regards,

Mohammad Reza Mahmoodi, Ph.D.

Academic Editor

PLOS ONE

Additional Editor Comments (optional):

Reviewers' comments:

Reviewer's Responses to Questions

**Comments to the Author**

1. If the authors have adequately addressed your comments raised in a previous round of review and you feel that this manuscript is now acceptable for publication, you may indicate that here to bypass the “Comments to the Author” section, enter your conflict of interest statement in the “Confidential to Editor” section, and submit your "Accept" recommendation.

Reviewer #1: (No Response)

Reviewer #2: All comments have been addressed

2. Is the manuscript technically sound, and do the data support the conclusions?

Reviewer #1: (No Response)

Reviewer #2: Yes

3. Has the statistical analysis been performed appropriately and rigorously? 

Reviewer #1: (No Response)

Reviewer #2: Yes

4. Have the authors made all data underlying the findings in their manuscript fully available?

Reviewer #1: (No Response)

Reviewer #2: Yes

5. Is the manuscript presented in an intelligible fashion and written in standard English?

Reviewer #1: (No Response)

Reviewer #2: Yes

6. Review Comments to the Author

Reviewer #1: Dear Authors

Many thanks for your contributions and response. all of my comments very good responded.

Best regards

Reviewer #2: (No Response)

7. PLOS authors have the option to publish the peer review history of their article (what does this mean?). If published, this will include your full peer review and any attached files.

Reviewer #1: No

Reviewer #2: No

---

## [Editor Report · Acceptance letter]

17 Oct 2023

PONE-D-23-01408R1 

Relationship between evacuation after the Great East Japan Earthquake and new-onset hyperuricemia: A 7-year prospective longitudinal study of the Fukushima Health Management Survey 

Dear Dr. Tanaka:

I'm pleased to inform you that your manuscript has been deemed suitable for publication in PLOS ONE. Congratulations! Your manuscript is now with our production department. 

Kind regards, 

on behalf of

Dr. Mohammad Reza Mahmoodi 

Academic Editor

PLOS ONE